# Cigarette Smoking Is Associated with Increased Risk of Malignant Gliomas: A Nationwide Population-Based Cohort Study

**DOI:** 10.3390/cancers12051343

**Published:** 2020-05-25

**Authors:** Stephen Ahn, Kyung-Do Han, Yong-Moon Park, Jung Min Bae, Sang Uk Kim, Sin-Soo Jeun, Seung Ho Yang

**Affiliations:** 1Department of Neurosurgery, Seoul St. Mary’s Hospital, College of Medicine, The Catholic University of Korea, Seoul 06591, Korea; nsstp@catholic.ac.kr (S.A.); ssjeun@catholic.ac.kr (S.-S.J.); 2Statistics and Actuarial Science, Soongsil University, Seoul 06978, Korea; hkd917@naver.com; 3Epidemiology Branch, National Institute of Environmental Health Sciences, National Institutes of Health, Research Triangle Park, NC 27709, USA; markparkjecos@gmail.com; 4Department of Dermatology, St. Vincent’s Hospital College of Medicine, The Catholic University of Korea, Seoul 16247, Korea; jminbae@gmail.com; 5MyoungJi St. Mary’s Hospital, Seoul 07417, Korea; tkddnr79@catholic.ac.kr; 6Department of Neurosurgery, St. Vincent’s Hospital, College of Medicine, The Catholic University of Korea, Suwon 16247, Korea

**Keywords:** glioma, smoking, risk factors, epidemiologic studies, population

## Abstract

The association between cigarette smoking and the risk of developing malignant glioma (MG) remains unclear. We aimed to evaluate this potential association in a large general population, using a well-established and validated longitudinal nationwide database. Using data from the Korean National Health Insurance System cohort, 9,811,768 people over 20 years old without any cancer history in 2009 were followed until the end of 2017. We documented 6100 MG cases (ICD-10 code C71) during the median follow-up period of 7.31 years. Current smokers had a higher risk of developing MG (HR = 1.22, CI: 1.13–1.32) compared with never-smokers, after adjusting for confounders. This association was stronger for those who smoked ≥ 20 cigarettes daily (HR = 1.50, CI: 1.36–1.64). Furthermore, having 30 or more pack-years of smoking over the course of one’s lifetime was associated with an increased risk of developing MG in a dose-dependent manner, compared with never-smokers (HR = 1.31, CI: 1.16–1.48 for 30–39 pack-years of smoking; HR = 1.36, CI: 1.17–1.59 for 40–49 pack-years of smoking; HR = 1.68; CI: 1.44–1.95 for ≥ 50 pack-years of smoking). These results suggest that cigarette smoking may be associated with developing MG. Further prospective studies could help elucidate this association.

## 1. Introduction

Malignant gliomas (MG) are the most common primary brain tumor in adults, with incidence rates of 5–10 per 100,000 person-years, and they usually have a devastating prognosis [1,2,3,4]. MG are generally classified into five main subtypes according to the World Health Organization (WHO) classification published in 2016 (glioblastoma, IDH-wild type; glioblastoma, IDH-mutant; astrocytoma, IDH-wild type; astrocytoma IDH-mutant; and oligodendroglioma, IDH-mutant and 1p19q co-deleted). In glioblastoma cases, which are the most common subtype and account for more than 70% of all gliomas, median survival is less than 1.5 years; median survival time of the other subtypes varies from 2 years to more than 10 years [4,5,6].

Cigarette smoking has already been established as one of the most significant risk factors for numerous cancers [7,8]. Clinical and animal studies have provided strong evidence that current smoking status is associated with increased risk of developing various cancers [9,10]. Moreover, risk increases proportionately to the cumulative lifetime amount of smoking in a dose-dependent manner [11,12].

While multifactorial factors, including environmental and genetic factors, are suggested to be associated with MG, the only proven risk factor is exposure to ionized radiation [13,14]. Etiology and susceptibility factors for MG remain largely unknown [13,14].

Recent biological findings have suggested a potential association between cigarette smoking and the risk of developing MG [15,16]. A number of cohort studies, case-control studies, and meta-analyses have explored this association epidemiologically, but results from these studies are inconclusive [17,18,19,20,21,22,23,24,25,26,27,28,29,30]. Among the studies that found an association were some limitations: two showed only positive associations in female populations [28,29], and a recent case-control study that included 4566 gliomas cases—the largest study to date—only found associations between cigarette smoking and death from gliomas, but not with risk for MG [19].

In this study, we evaluated the potential association between cigarette smoking and MG risk in a large sample, using a nationwide population database of Koreans. The Korea National Health Insurance Service (NHIS), which is a mandatory health insurance system covering 97% of all Koreans, has recently established a patient cohort. A number of previous studies have generated powerful results for identifying cancer risks by using this large nationwide database [31,32,33].

## 2. Results

### 2.1. Characteristics of the Study Population

Among the 9,811,768 patients in our study population, 5,820,623 (59.32%) were never-smokers, 1,400,124 (14.27%) were former smokers, and 2,591,021 (26.41%) were current smokers. Never-smokers and former smokers were significantly older than current smokers (*p* < 0.001). Being male and related factors such as height, weight, waist circumference, and heavy drinking were significantly associated with former or current smokers (*p* < 0.001). Detailed characteristics of populations were described in Table 1.

### 2.2. Incidence Rates and Risk of Developing MG According to Smoking Status

A total of 6100 MGs developed during the median follow-up period of 7.31 years and 71,280,380 person-years. MG incidence rates per 100,000 person-years were 8.80 in never-smokers, 8.91 in former smokers, and 7.81 in current smokers (Table 2). Because the mean age of current smokers was significantly younger than that of never and former smokers, and age is one of the most significant risk factors for MG, we analyzed MG incidence rates according to age by decade. After analyzing, we found that current smokers showed significantly increased incidence rates of MG than never and former smokers, in populations 50 years and older (Appendix A). In adjusted models to correct for age and other possible confounding factors, current smokers showed a significantly higher risk of developing MG than former never-smokers (HR 1.19, CI 1.10–1.28 and HR 1.22, CI 1.13–1.32 in model 1 and 2, respectively) (Table 2).

### 2.3. Risk of Developing MG According to Smoking Amount and Duration

We analyzed the association between MG risk and per-day number of cigarettes smoked (Table 3). Among current smokers, there was a significantly higher risk of developing MG in people who smoked >20 cigarettes per day compared with never-smokers (HR 1.44, CI 1.32–1.58 & HR 1.50, CI 1.36–1.64, in model 1 and 2, respectively). Additionally, former smokers who used to smoke >20 cigarettes per day showed an increased risk of developing MG, relative to never-smokers (HR 1.15, CI 1.00–1.30 in model 2). Former smokers who smoked <20 cigarettes per day showed no significant differences in MG risk compared with never-smokers.

We also analyzed the associations between developing MG and smoking duration (Table 3). In model 2, current smokers had significantly increased risk of developing MG, regardless of smoking duration. Former smokers had no increased risk of developing MG compared with never-smokers, regardless of smoking duration.

### 2.4. MG Risk According to Cumulative Lifetime Amount of Smoking

We found a significant dose-dependent association between cumulative lifetime amount of smoking and risk of developing MG (Figure 1a). People who ever smoked >30 pack-years in their lifetime had a significantly increased risk of developing MG compared with never-smokers (HR 1.31, CI 1.16–1.48). Among ever-smokers, the risk of developing MG increased in proportion to the cumulative amount of lifetime smoking: the highest risk of developing MG was among people who ever smoked >50 pack-years (HR 1.68, CI 1.44–1.95), followed by people who ever smoked 40–49 pack-years (HR 1.36, CI 1.17–1.59), and then by people who ever smoked 30–39 pack-years (HR 1.31, CI 1.16–1.48) (Figure 1b).

### 2.5. Subgroup Analyses According to Age and Sex

We analyzed subgroup according to age and sex in model 2. While current smokers older than 40 years had an increased risk of developing MG, younger current smokers less than 40 years showed no associations with developing MG (Table 4). Male smokers showed increased risks of developing MG, however, female smokers did not show any significant differences in developing MG compared to never smokers (Table 4).

All data are reported as hazard ratios with 95% confidence intervals from adjusted model 2, which is adjusted for age, sex, alcohol consumption, body-mass index, income level, and exercise level.

## 3. Discussion

In this nationwide population-based cohort study, we examined the risk of developing MG according to current smoking status, the amount of cigarettes smoked per day, smoking duration, and the cumulative amount of smoking exposure over the lifetime. We counted newly diagnosed malignant gliomas among a large general population of Koreans (9,811,768) from January 2009 to December 2017 and calculated the overall MG incidence rate: 8.56/100,000 person-years. This estimate is consistent with recent epidemiological glioma research [4,34]. We found a graded association between the cumulative amount of lifetime smoking and risk of MG among people who smoked ≥30 pack-years.

To the best of our knowledge, this study included the largest number of glioma cases (*n* = 6100) among studies that evaluated the association between smoking and MG risk in adults. We were able to enroll a sufficiently large sample population (~10 million people) with complete medical information, that was updated annually using the NHIS database from South Korea. This large and powerful database has already been validated by our previous studies [31,32,33,35]. Prior to this, the largest study was a case-control study including 4556 glioma cases [19], that investigated cigarette smoking and death from glioma, and found that smoking increased the risk of glioma death by 11% (OR = 1.11, CI: 1.03–1.21). All previously published studies that evaluated potential associations between smoking and MG risk included fewer than 1000 glioma cases [21,22,23,24,26,27,28,29,30]. Among them, a cohort study including 89,935 Canadian women found an association between smoking and glioma incidence among women who smoked for more than 20 years (OR 1.51, CI 0.97–2.34), smoked more than 20 cigarettes per day (OR 1.44, CI 0.90–2.31), or smoked more than 15 pack-years (OR 1.50, CI 0.92–2.44); however, none of these trends were significant [28]. MG is a relatively rare disease, with incidence rates lower than 10/100,000, thus, a very large population sample is needed to achieve meaningful statistical power. This could explain why most studies yielded negative results, while our study showed a prominent association between smoking and glioma risks.

We estimated the cumulative amount of lifetime smoking by calculating people’s average pack-years, which is considered the most reliable parameter for analyzing the effects of smoking on the human body [11]. Using these data, we identified a graded association between the cumulative amount of lifetime smoking and MG risk for people who smoked 30 or more pack-years. The risk of developing MG increased in a dose-dependent manner, the more pack-years smoked (31% increase among people who ever smoked 30–39 pack-years, 36% among people who ever smoked 40–49 pack years, 68% among people who ever smoked >50 pack-years). Unfortunately, we did not find a dose-dependent relationship between risk of MG and smoking duration, although current smokers had a significantly increased risk of MG, regardless of smoking duration. Moreover, former smokers who smoked less than 19 cigarettes per day or smoked for less than 29 years showed decreased risk compared with current and never smokers. This finding may suggest that cessation of smoking or reducing the amount of smoking per day could decrease the risk of developing MG like other types of cancer [34,36]. Further studies are needed to elucidate the effect of smoking cessation on glioma. Given our findings, physicians should consider discussing smoking cessation or reduction with their long-term heavy smokers during clinic appointments, to reduce their risk of developing MG.

Studies that have focused on pathophysiological mechanisms have provided prominent biological evidence of an association between smoking and glioma risk. A recent biological study including more than 2000 human tissues samples from various cancers showed multiple distinct mutational signatures induced by smoking, both in tissues directly exposed to smoking and tissues indirectly exposed to smoking [10]. Furthermore, nicotine, a major ingredient in cigarettes, was found to enhance the proliferation, migration, and radio-resistance of human malignant glioma cells [15]. Other recent studies showed that the blood-brain barrier, which is a unique anatomical structure that acts as a gatekeeper against hazardous agents, could be breached by cigarette smoking via the impairment of endothelial tight junctions [16,37,38]. Thus, smoking may facilitate various environmental carcinogens in penetrating the brain and damaging the brain [7,10].

Our findings should be considered in light of several limitations. First, some data were acquired from self-reported questionnaires, which may introduce biases to our dataset. Second, although we considered many factors that potentially influenced the hazard ratios for the MG risk, several undefined factors were not included for adjustment. Third, although MG is comprised of heterogeneous subtypes, some of which are more frequent in the younger population than in the older population, analysis according to subtype was not possible in our study design. Further studies are needed to analyze the incidence and risk factors according to histologic and molecular subgroup of glioma. Fourth, we verified the accuracy of our methodology to identify gliomas by retrospectively reviewing electronic medical records at a tertiary referral hospital in Korea. However, these results from a tertiary hospital may not be generalizable to other hospitals in Korea. Lastly, we only found an association between smoking and risk of developing MG among the male patients in our sample. In contrast, previous studies found associations among female populations [19,28,29]. This discrepancy may be because the proportion of never-smokers in our female subset was >94%, which leaves a significantly lower proportion of female smokers to analyze compared with other female populations.

## 4. Materials and Methods

### 4.1. Ethical Considerations

This study was conducted in accordance with the ethical standards of the 1964 Declaration of Helsinki. The Institutional Review Board at our institution approved the study design (ethical code: KC18ZESI0648, permission date: 23 October 2018). To protect patients’ private information, all individual data were anonymized. Owing to the retrospective nature of the study, the informed consent requirement was waived.

### 4.2. Database Source

This retrospective national-wide population-based study was performed using the National Health Insurance Service (NHIS) database from Korea, which is a mandatory health insurance system operated by the Korean government that covers almost 97% of Koreans (approximately 50 million people). The NHIS database includes medical information from each patient regarding demographics, clinical diagnoses, prescribed medications including chemotherapy, surgical procedures, and radiotherapy. Furthermore, through a routine national health examination that is provided by the NHIS, either for all enrolled adults >40 years old at least every two years, or for any workers at a company >20 years old, physical measurements, including height, weight, body-mass index (BMI), medical history, family history, social-behavioral history including cigarette smoking, alcohol consumption, and physical activity were obtained from self-reported questionnaire records.

### 4.3. Study Population

We reviewed the records from the NHIS database for all people who were >20 years old by 2009. Because NHIS limits its database size to 10 million people due to the personal information protection act, 9,811,768 people were identified after excluding people with any cancer history and incomplete medical data. Among this dataset, a total of 6100 malignant gliomas were identified from January 2009 to December 2017. The mean follow-up periods for enrolled individuals were 7.31 years and 71,280,381 person-years. The overall flow for patient enrollment is illustrated in Figure 2.

### 4.4. Definitions

The medical code “C71” represents MG in the brain according to the International Classification of Disease, Tenth Revision (ICD-10), and includes all malignant gliomas, such as diffuse astrocytoma, anaplastic astrocytoma, ependymoma, anaplastic ependymoma, oligodendroglioma, anaplastic oligodendroglioma, and glioblastoma multiforme. Because all C71 patients received the additional cost coverage service from the NHIS for rare and incurable diseases, called the “benefit extension policy for rare incurable disease (BEP)”, we defined MG for patients who were both diagnosed with C71 and registered into the benefit extension policy to ensure accurate categorization of the study population. To further verify the accuracy of our method to identify MG, we retrospectively reviewed the electronic medical records at Seoul St. Mary’s Hospital, a tertiary referral hospital in Korea. After recruiting patients who visited this hospital between 2014 and 2018, we analyzed individual patient medical records for people who fit our definition of MG. We found a total of 220 patients who were radiologically or pathologically confirmed to have MG.

Smoking status was classified according to self-report questionnaire records. Current smokers were defined as people who smoked more than five packs (a total of 100 cigarettes) over their lifetime and continued to smoke; former smokers were defined as people who smoked more than five packs (a total of 100 cigarettes) over their lifetime, but who quit smoking prior to completing the questionnaire; and never-smokers were defined as people who had ever smoked five packs or fewer. The total duration of smoking years and the average daily number of cigarettes smoked (number of cigarettes per day) were included in the self-questionnaires. We calculated the cumulative amount of smoking exposure throughout lifetimes (i.e., pack-years) by multiplying the average per-day cigarette consumption by the smoking period. For alcohol consumption, heavy drinkers were defined as individuals who consumed an average of ≥30 g of alcohol per day. For physical activity, regular exercise was defined as intensive physical activity with faster-than-normal breathing for ≥20 min at a time more than three days per week, and moderate physical activity with slightly faster-than-normal breathing ≥30 min at least five days per week. Socio-economic status was classified based on yearly payments to the NHIS, according to income level.

### 4.5. Statistical Analyses

Data are expressed as means ± standard deviations for continuous variables and as proportions for categorical variables. The one-way analysis of variance (ANOVA) test was used to compare differences between continuous variables, and the chi-square test was used to compare differences between categorical variables. The incidence rates for MG were calculated and expressed as the number of events per 100,000 person-years. We fit a model adjusted for the potential confounders of age and sex (model 1), and we fit another model that included the potential confounders from model 1, but also included alcohol consumption, body-mass index, income level, and exercise level (model 2). The cumulative incidences rates for MG were compared between groups using the Kaplan–Meier method and the log-rank test. Cox proportional hazards models were used to analyze the adjusted risk of malignant glioma development, based on smoking status and daily amount or smoking duration; the results are expressed as hazard ratios (HR) with 95% confidence intervals (CI). A *p*-value < 0.05 was considered statistically significant. Statistical analyses were performed using SAS version 9.4 (SAS Institute, Cary, NC, USA).

## 5. Conclusions

This nationwide population-based study revealed that cigarette smoking may be significantly associated with an increased risk of developing MG. The MG incidence rate increased in proportion to the cumulative lifetime smoking amount among people who smoked more than 30 pack-years. Further prospective studies are needed to evaluate these associations, and preclinical studies should also be performed to define the biological mechanisms linking smoking and MG development in the brain, with focus on the unique anatomy of the blood-brain barrier.

## Figures and Tables

**Figure 1 cancers-12-01343-f001:**
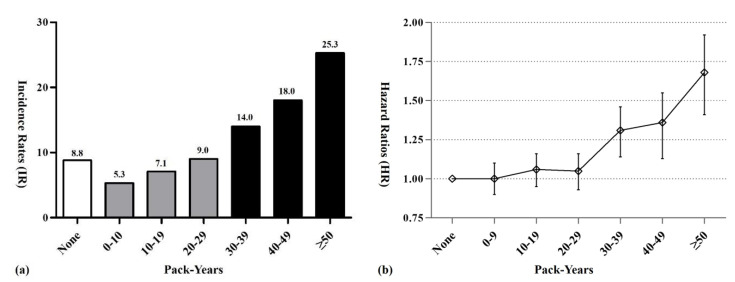
Incidence rates (**a**) and risk of developing (**b**) malignant glioma, according to cumulative amount of lifetime smoking.

**Figure 2 cancers-12-01343-f002:**
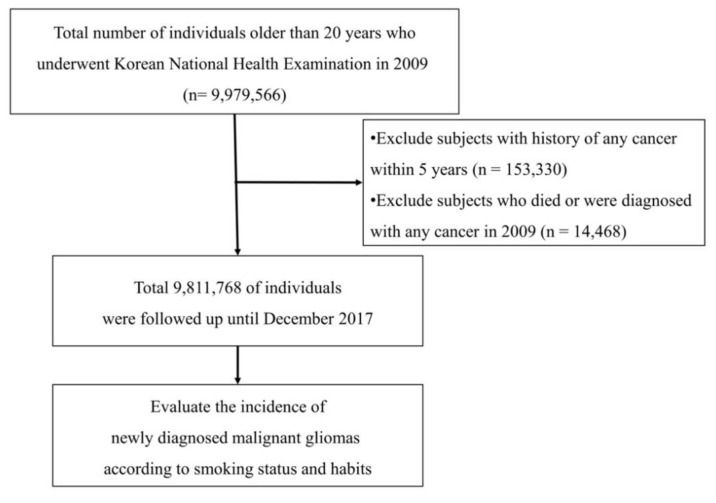
Flow of the study design.

**Table 1 cancers-12-01343-t001:** Baseline characteristics of the study population according to smoking status.

Characteristic Title	Never-Smokers	Former Smokers	Current Smokers	*p*-Value
	*n* = 5,820,623 (59.32)	*n* = 1,400,124 (14.27)	*n* = 2,591,021 (26.41)	
Mean age, years ^a^	48.47 ± 14.55	48.67 ± 13.03	42.66 ± 12.49	<0.001
20–39, *n* (%)	1,576,423 (27.08)	364,065 (26.00)	1,173,409 (45.29)	
40–64, *n* (%)	3,336,188 (57.32)	854,543 (61.03)	1,253,281 (48.37)	
≥65, *n* (%)	908,012 (15.6)	181,516 (12.96)	164,331 (6.34)	
Male	1,634,168 (28.08)	1,316,955 (94.06)	2,437,271 (94.07)	<0.001
Height, cm ^a^	160.03 ± 8.55	169.01 ± 6.71	169.98 ± 7.01	<0.001
Weight, cm ^a^	60.26 ± 10.52	69.62 ± 10.22	69.25 ± 11.33	<0.001
Waist circumference, cm ^a^	78.27 ± 9.16	83.95 ± 7.87	82.61 ± 8.24	<0.001
BMI, kg/m^2 a^	23.47 ± 3.25	24.32 ± 2.91	23.9 ± 3.21	<0.001
Urban residence, *n* (%)	3,123,516 (53.66)	739,082 (52.79)	1,434,412 (55.36)	<0.001
Heavy drinker ^b^, *n* (%)	117,516 (2.02)	161,722 (11.55)	399,344 (15.41)	<0.001
Regular exercise ^c^, *n* (%)	2,705,616 (46.48)	905,008 (64.64)	1,436,112 (55.43)	<0.001
Diabetes mellitus, *n* (%)	469,259 (8.06)	154,318 (11.02)	221,970 (8.57)	<0.001
Hypertension, *n* (%)	1,507,526 (25.9)	440,210 (31.44)	560,252 (21.62)	<0.001
Dyslipidemia, *n* (%)	1,102,507 (18.94)	281,650 (20.12)	403,142 (15.56)	<0.001
Previous ischemic heart disease, *n* (%)	112,688 (3.12)	45,101 (4.78)	35,888 (2.19)	<0.001
Previous cerebral stroke, *n* (%)	50,943 (1.41)	26,027 (2.76)	22,864 (1.39)	<0.001
Mean amounts of cigarette smoked per day	0	16.0 ± 9.1	16.0 ± 7.3	
Mean smoking duration	0	16.2 ± 10.6	18.8 ± 11.1	
Mean pack-years	0	14.6 ± 14.6	16.0 ± 13.2	
Hemoglobin, g/dL	13.33 ± 1.49	14.69 ± 1.25	14.97 ± 1.26	<0.001

BMI, body-mass index; BP, blood pressure; *n*, number; ^a^ Mean ± standard deviation; ^b^ Defined as a person who consumed >30 g of alcohol per day on average; ^c^ Defined as intensive physical activity of >3 days per week or moderate physical activity of >5 days per a week.

**Table 2 cancers-12-01343-t002:** Incidence rates and risk of malignant gliomas, according to smoking status.

Smoking Status	Total, *n*	MG Events, *n*	Person-Years	MG Incidence Rate *	Crude HR (95% CI)	^a^ Model 1 HR (95% CI)	^b^ Model 2 HR (95% CI)
Never-smokers	5,820,623	3733	42,405,897	8.80	1 (reference)	1 (reference)	1 (reference)
Former smokers	1,400,124	904	10,143,343	8.91	1.01 (0.94–1.09)	0.98 (0.90–1.07)	1.00 (0.92–1.09)
Current smokers	2,591,021	1463	18,731,140	7.81	0.89 (0.84–0.94)	1.19 (1.10–1.28)	1.22 (1.13–1.32)

CI, confidence interval; HR, hazard ratio; MG, malignant glioma; *n*, number; * per 100,000 person-years; ^a^ Model 1: adjusted for age and sex; ^b^ Model 2: adjusted for model 1 plus alcohol consumption, body-mass index, income level, and exercise level.

**Table 3 cancers-12-01343-t003:** Risk of developing malignant glioma according to smoking amount and duration.

Smoking Status	Total Number, *n* (% of Proportion)	MG Events, *n*	Person-Years	MG Incidence Rate *	Crude HR (95% CI)	^a^ Model 1 HR (95% CI)	^b^ Model 2 HR (95% CI)
Never-smokers	5,820,623 (59.3)	3,733	42,513,670	8.80	1 (reference)	1 (reference)	1 (reference)
Former smokers							
Cigarettes smoked per day							
<10	364,712 (3.7)	155	2,620,568	5.72	0.84 (0.71–1.00)	0.95 (0.80–1.12)	0.97 (0.81–1.14)
10–19	818,051 (8.3)	467	5,949,887	7.85	0.88 (0.78–0.99)	0.92 (0.83–1.03)	0.94 (0.85–1.05)
≥20	217,361 (2.2)	282	1,528,462	18.45	1.20 (1.10–1.32)	1.12 (0.98–1.28)	1.15 (1.00–1.30)
Smoking duration (years)							
<10	251,965 (2.6)	136	1,832,908	5.72	0.66 (0.56–0.78)	0.91 (0.77–1.09)	0.93 (0.78–1.11)
10–29	534,103 (5.4)	300	3,880,899	7.85	0.88 (0.81–0.98)	0.93 (0.82–1.05)	0.95 (0.84–1.08)
≥30	614,056 (6.3)	468	4,429,536	18.45	2.10 (1.86–2.37)	1.05 (0.94–1.17)	1.07 (0.96–1.19)
Current smokers							
Cigarettes smoked per day							
<10	426,590 (4.3)	129	3,051,709	4.03	0.97 (0.84–1.12)	1.02 (0.85–1.23)	1.05 (0.87–1.26)
10–19	1,631,714 (16.6)	604	11,868,42	5.09	0.75 (0.58–0.82)	0.98 (0.88–1.08)	1.00 (0.90–1.10)
≥20	532,717 (5.4)	730	3,747,653	19.48	0.99 (0.92–1.08)	1.45 (1.32–1.59)	1.50 (1.37–1.65)
Smoking duration (years)							
<10	323,511 (3.3)	199	2,331,871	4.03	0.47 (0.40–0.56)	1.21 (1.04–1.40)	1.25 (1.07–1.44)
10–29	1,089,332 (11.1)	518	7,895,617	5.09	0.58 (0.53–0.63)	1.10 (0.99–1.21)	1.13 (1.02–1.25)
≥30	1,178,178 (12.0)	746	8,503,651	19.48	2.22 (2.05–2.40)	1.26 (1.15–1.38)	1.29 (1.18–1.42)

CI, confidence interval; HR, hazard ratio; MG, malignant glioma; *n*, number, * per 100,000 person-years, ^a^ Model 1: adjusted for age and sex, ^b^ Model 2: adjusted for model 1 plus alcohol consumption, body-mass index, income level, and exercise level.

**Table 4 cancers-12-01343-t004:** Subgroup analyses of risk of malignant glioma, according to age and sex.

Smoking Status	Age			Sex	
	20–39	40–64	≥65	Male	Female
Never-smokers	1 (reference)	1 (reference)	1 (reference)	1 (reference)	1 (reference)
Former smokers	1.20 (0.93–1.55)	1.03 (0.92–1.17)	0.92 (0.78–1.06)	1.03 (0.94–1.13)	0.85 (0.60–1.21)
Current smokers	1.08 (0.88–1.32)	1.27 (1.14–1.41)	1.37 (1.20–1.50)	1.30 (1.12–1.41)	1.06 (0.84–1.32)

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
