# Peer review of "Cigarette Smoking Is Associated with Increased Risk of Malignant Gliomas: A Nationwide Population-Based Cohort Study"

_cancers, 2020, doi:10.3390/cancers12051343_

Round 1

Reviewer 1 Report

The authors present a study about the influence of smoking and the risk to develop gliomas. The risk factores for gliomas are unclear and I think this study is interesting.   However there some points that I would like to be clarified:   1) Introduction shoud be improved namely relatect with gliomas risk factors known, such as radiation. 2) The introduction should mencionat more about gliomas namely  new classification, where there are 5 molecular subgroups. 3) Regarding results, the subtype of gliomas are importante we know that some subtypes are more frequente in young population and other in older population. The incidence and risk should be done according with molecular subgourp of gliomas. 4) Figure 1 is inverted 

Author Response

Revision Letter

Title: Cigarette smoking is associated with increased risk of malignant gliomas: A nationwide population-based cohort study

Manuscript number: Cancers-781770

Dear editor-in-chief and editors

Thank you for reviewer’s opinions.

We respond to issues mentioned in the reviewer’s comments as follows.

Our responses are presented in green font, and changes in the article are denoted in red font.

We would be grateful if revised manuscript could be considered for publication.

Best regards,

Seung Ho Yang, M.D.

Department of Neurosurgery, St. Vincent’s Hospital, The Catholic University of Korea,

93 Jungbudaero, Paldal-gu, Suwon, Korea, 16247

Tel: +82.31-249-8303; Fax: +82.31-245-5208

Responses to Comments from the editors and reviewers

Reviewer #1

The authors present a study about the influence of smoking and the risk to develop gliomas. The risk factors for gliomas are unclear and I think this study is interesting.   However there some points that I would like to be clarified:  

 Response: We appreciate your comments and have revised our manuscript as suggested.

1) Introduction should be improved namely related with gliomas risk factors known, such as radiation.

 Response: We agree with your comment and have added a few sentences to clarify glioma risk factors.

 Change to text: (in 1. Introduction)

(Revised Manuscript file: page 2, 3rd paragraph, 1~3rd lines)

“While multifactorial factors, including environmental and genetic factors, are suggested to be associated with MG, the only proven risk factor is exposure to ionized radiation. Etiology and susceptibility factors for MG remain largely unknown.”

2) The introduction should mention more about gliomas namely new classification, where there are 5 molecular subgroups.

 Response: We agree with your comment and have changed and added a few sentences to clarify gliomas and the new classification.

 Change to text: (in 1. Introduction)

(Revised Manuscript file: page 1, 1st paragraph, 2nd lines to page 2, 1st paragraph, 1st~2nd lines)

“MG are generally classified into five main subtypes according to the World Health Organization (WHO) classification published in 2016 (glioblastoma, IDH-wild type; glioblastoma, IDH-mutant; astrocytoma, IDH-wild type; astrocytoma IDH-mutant; and oligodendroglioma, IDH-mutant and 1p19q co-deleted). In glioblastoma cases, which are the most common subtype and account for more than 70% of all gliomas, median survival is less than 1.5 years; median survival time of the other subtypes vary from 2 years to more than 10 years.”

3) Regarding results, the subtype of gliomas is important we know that some subtypes are more frequent in young population and other in older population. The incidence and risk should be done according with molecular subgroup of gliomas.

 Response: We agree with your comments. Unfortunately, obtaining histologic and molecular subgroups of glioma from the NHIS database was not possible. We have corrected and added a few sentences to clarify this limitation in the Discussion.

 Change to text: (in 3. Discussion)

(Revised Manuscript file: page 6, 4th paragraph, 4-9th lines)

“Although MG is comprised of heterogeneous subtypes, some of which are more frequent in the younger population than in the older population, analysis according to subtype was not possible in our study design. Further studies are needed to analyze the incidence and risk factors according to histologic and molecular subgroup of glioma.”

4) Figure 1 is inverted.

 Response: We agree with your comment and have corrected the figure.

 Change: (in 2. Results)

(Revised Manuscript file: page 5, 1st paragraph)

*We also attached the revised manuscript.

Reviewer 2 Report

Ahn et al present data in this manuscript that associate smoking with increased risk of malignant glioma in the Korea population. They show that by controlling for age, heavy smokers have about 20% higher risk of developing glioma, and this risk is mainly for male rather than female. Furthermore, they show that this risk is also dosage dependent – more smoking is associated with increased risk. These results seem to agree with other studies, although it is surprising that no significant association was found in women.

Overall, this is a well conducted study. The conclusions are well supported by the data. I do not have major criticisms. A minor issue is regarding figure 1b – it is upside down.

Author Response

Revision Letter

Title: Cigarette smoking is associated with increased risk of malignant gliomas: A nationwide population-based cohort study

Manuscript number: Cancers-781770

Dear editor-in-chief and editors

Thank you for reviewer’s opinions.

We respond to issues mentioned in the reviewer’s comments as follows.

Our responses are presented in green font, and changes in the article are denoted in red font.

We would be grateful if revised manuscript could be considered for publication.

Best regards,

Seung Ho Yang, M.D.

Department of Neurosurgery, St. Vincent’s Hospital, The Catholic University of Korea,

93 Jungbudaero, Paldal-gu, Suwon, Korea, 16247

Tel: +82.31-249-8303; Fax: +82.31-245-5208

Responses to Comments from the editors and reviewers

Reviewer #2

Ahn et al present data in this manuscript that associate smoking with increased risk of malignant glioma in the Korea population. They show that by controlling for age, heavy smokers have about 20% higher risk of developing glioma, and this risk is mainly for male rather than female. Furthermore, they show that this risk is also dosage dependent – more smoking is associated with increased risk. These results seem to agree with other studies, although it is surprising that no significant association was found in women. Overall, this is a well conducted study. The conclusions are well supported by the data. I do not have major criticisms.

 Response: We appreciate your comments and have revised our manuscript as suggested.

1) A minor issue is regarding figure 1b – it is upside down.

 Response: We agree with your comment and have corrected the figure.

 Changes: (in 2. Results)

(Revised Manuscript file: page 5, 1st paragraph)

* We also attached the revised manuscript.

Reviewer 3 Report

1. In Table 3, for Current Smokers in Model 1 and Model 2, there seems to be a decrease in risk from Smoking Duration from 10-29 years compared to the Current Smokers who smoked < 10 years and those who smoked ≥ 30 years.  Please explain these differences in further detail.

2. The definition of MG is quite broad under Section 4.4, and appears to include grade 2 diffuse astrocytomas, grade 2 oligodendrogliomas, and grade 2 ependymomas.  These histologic tumors are not typically considered "malignant" gliomas relative to the anaplastic gliomas and glioblastomas.  Either the study should include all gliomas grade 2 or above, or the analyses should be restricted to true malignant gliomas, as it's possible that the association with smoking and true malignant gliomas may be stronger.  Also, this may minimize the age bias in which most of the current smokers are younger, and thus more likely to develop lower grade gliomas than MG.

3. Figure 1b is vertically flipped and should be corrected.

4. Under the Discussion section (line 180) the word "Forth" is misspelled, and should be corrected to "Fourth".

Author Response

Revision Letter

Title: Cigarette smoking is associated with increased risk of malignant gliomas: A nationwide population-based cohort study

Manuscript number: Cancers-781770

Dear editor-in-chief and editors

Thank you for reviewer’s opinions.

We respond to issues mentioned in the reviewer’s comments as follows.

Our responses are presented in green font, and changes in the article are denoted in red font.

We would be grateful if revised manuscript could be considered for publication.

Best regards,

Seung Ho Yang, M.D.

Department of Neurosurgery, St. Vincent’s Hospital, The Catholic University of Korea,

93 Jungbudaero, Paldal-gu, Suwon, Korea, 16247

Tel: +82.31-249-8303; Fax: +82.31-245-5208

Responses to Comments from the editors and reviewers

Reviewer #3

1) In Table 3, for Current Smokers in Model 1 and Model 2, there seems to be a decrease in risk from Smoking Duration from 10-29 years compared to the Current Smokers who smoked < 10 years and those who smoked ≥ 30 years.  Please explain these differences in further detail.

 Response: We agree with your comment. We did our best to explain these differences in detail. The effect of smoking on cancer incidence can be measured using cigarettes smoked per day, duration of smoking, or pack-years, which combines the two factors. We determined that pack-years was the most reliable parameter and suggested that the risk of MG increased in proportion to pack-years of smoking over a lifetime. However, as you mentioned, there was some discordance in terms of duration of smoking. We noticed similar differences in a recent study published in Neuro-Oncology (Hou, Lei, et al. "Smoking and adult glioma: a population-based case-control study in China." Neuro-oncology 18.1 (2015): 105-113.). Smokers who smoked 10-19 cigarettes daily showed decreased risk compared to those who smoked < 10 cigarettes daily, and female smokers who smoked for 20-29 years showed decreased risk compared to those who smoked < 20 years.  The authors did not explain the reason for these differences. We assume that using only smoking duration or cigarettes smoked per day is associated with confusing results that can be avoided with use of pack-years.

 Change to text: (in 3. Discussion)

(Revised Manuscript file: page 6, 2th paragraph, 7-9th lines)

“Unfortunately, we did not find a dose-dependent response between risk of MG and smoking duration, although current smokers had significantly increased risk of MG regardless of smoking duration.”

2) The definition of MG is quite broad under Section 4.4, and appears to include grade 2 diffuse astrocytomas, grade 2 oligodendrogliomas, and grade 2 ependymomas.  These histologic tumors are not typically considered "malignant" gliomas relative to the anaplastic gliomas and glioblastomas.  Either the study should include all gliomas grade 2 or above, or the analyses should be restricted to true malignant gliomas, as it's possible that the association with smoking and true malignant gliomas may be stronger.  Also, this may minimize the age bias in which most of the current smokers are younger, and thus more likely to develop lower grade gliomas than MG.

 Response: We appreciate and agree with your comment. There is a question whether grade II glioma should be considered benign or malignant. Some grade II gliomas are slowly growing and show favorable survival outcomes; however, a reference from American Brain Tumor Association (https://www.abta.org/wp-content/uploads/2018/03/low-grade-astrocytoma.pdf) suggest that they should not be considered benign. Also, recent studies showed that even grade II glioma with specific molecular features such as IDH mutation had prognosis as poor as that of grade IV glioblastoma (Brat, Daniel J., et al. "cIMPACT-NOW update 3: recommended diagnostic criteria for “Diffuse astrocytic glioma, IDH-wildtype, with molecular features of glioblastoma, WHO grade IV”." Acta neuropathologica 136.5 (2018): 805-810).  We agree with the age bias you mentioned and have added a few sentences to explain this potential bias.

 Change to text: (in 3. Discussion)

(Revised Manuscript file: page 6, 4th paragraph, 4-9th lines)

“Although MG is comprised of heterogeneous subtypes, some of which are more frequent in the younger population than in the older population, analysis according to subtype was not possible in our study design. Further studies are needed to analyze the incidence and risk factors according to histologic and molecular subgroup of glioma.”

3) Figure 1b is vertically flipped and should be corrected.

 Response: We agree with your comment and have corrected the figure.

 Change: (in 2. Results)

(Revised Manuscript file: page 5, 1st paragraph)

*We also attached the revised manuscript.

4) Under the Discussion section (line 180) the word "Forth" is misspelled, and should be corrected to "Fourth".

 Response: We agree with your comment and have corrected the word.

 Change to text: (in 3. Discussion)

(Revised Manuscript file: page 6, 4th paragraph, 6th line)

“Fourth, …”
